# The effect of BC on aerosol-boundary layer feedback: Potential implications for urban pollution episodes

Jessica Slater[1,2], Hugh Coe[1], Gordon McFiggans[1], Juha Tonttila[3], and Sami Romakkaniemi[3]

[1]Stockholm Environment Institute, Department of Environment and Geography, University of York, York, UK
[2]Centre for Atmospheric Science, School of Earth and Environmental Sciences, University of Manchester, Manchester, UK
[3]Finnish Meteorological Institute, Atmospheric Research Centre of Eastern Finland, Kuopio, Finland

**Correspondence:** Hugh Coe (hugh.coe@manchester.ac.uk)

**Abstract.**

Beijing suffers from poor air quality particularly during wintertime haze episodes when concentrations of $PM_{2.5}$ can peak at > 400 ug/m$^3$. Black carbon (BC), an aerosol which strongly absorbs solar radiation can make up to 10 % of $PM_{2.5}$ in Beijing. BC is of interest due to its climatic and health impacts. BC has also been found to impact planetary boundary layer (PBL) meteorology. Through interacting with radiation and altering the thermal profile of the lower atmosphere, BC can either suppress or enhance PBL development depending on the properties and altitude of the BC layer.

Previous research assessing the impact of BC on PBL meteorology has been investigated through the use of regional models which are limited both by resolution and the chosen boundary layer schemes. In this work, we apply a high-resolution model (UCLALES-SALSA) that couples an aerosol and radiative transfer model with large-eddy simulation (LES) to quantify the impact of BC at different altitudes on PBL dynamics using conditions from a specific haze episode which occurred from 1st-4th Dec 2016 in Beijing. Results presented in this paper quantify the heating rate of BC at various altitudes to be between 0.01 and 0.016 K/h per $\mu$g/m$^3$ of BC, increasing with altitude but decreasing around PBL top. Through utilising a high resolution model which explicitly calculates turbulent dynamics, this paper showcases the impact of BC on PBL dynamics both within and above the PBL. These results show that BC within the PBL increases maximum PBL height by 0.4 % but that the same loading of BC above the PBL can suppress PBL height by 6.5 %. Furthermore, when BC is present throughout the column the impact of BC suppressing PBL development is further maximised, with BC causing a 17 % decrease in maximum PBL height compared to only scattering aerosols. Assessing the impact of these opposite effects, in this paper, we present a mechanism through which BC may play a prominent role in the intensity and longevity of Beijing's pollution episodes.

## 1 Introduction

Beijing, a megacity situated in the North China Plain, experiences extremely poor air quality. Typically in Beijing wintertime, heavy pollution episodes termed 'haze' envelop the city and concentrations of $PM_{2.5}$ (particulate matter with a diameter < 2.5 $\mu$m) frequently exceed the recommended World Health Organization exposure limits (10 $\mu$g/m$^3$ annual average and 25 $\mu$g/m$^3$ 24-hour average) (WHO, 2006). Poor air quality has been linked to various respiratory and cardiovascular diseases as well

as neurodegenerative diseases such as Parkinson's and dementia (Yang et al., 2013; Lelieveld et al., 2015; Chen et al., 2017). Improving air quality is therefore a public health priority for the Chinese government. However, despite policy interventions which have improved annual average air quality in Beijing over the past decade, heavy pollution episodes are still a major issue (Chan and Yao, 2008). Black carbon (BC), primarily emitted through incomplete combustion, is a strongly absorbing aerosol present in high concentrations in Chinese megacities (Fu and Chen, 2016). In Beijing, BC can contribute up to 10 % of total particulate matter (PM) during polluted periods (Liu et al., 2016). Major sources of BC in Beijing are: traffic, biomass burning and coal combustion for both residential and industrial use (Streets et al., 2001).

BC is of interest globally due to its climatic and health impacts. As a short lived climate pollutant which strongly absorbs radiation across the shortwave (SW) spectrum, BC can directly cause atmospheric warming and is considered to be the largest anthropogenic contributor to global warming after carbon dioxide ($CO_2$) (Bond et al., 2013). Furthermore, due to its relatively short lifetime in the atmosphere (days) compared to $CO_2$ (years), reducing concentrations of BC in the atmosphere could have a rapid impact on global temperatures, with the added benefit of improving air quality for human health. The global direct radiative forcing of BC at top of atmosphere (TOA) is estimated to be between 0.2-1.2 $W/m^2$ (Ramanathan and Carmichael, 2008; Bond et al., 2013). Calculating the global direct radiative forcing effect (DRE) of BC is complicated by its spatial heterogeneity, with a higher effect (up to 10 $W/m^2$) in heavily polluted urban areas where BC concentrations are significantly higher (Ferrero et al., 2014; Li and Han, 2016). Furthermore, the DRE of BC is significantly affected by its source, vertical distribution, atmospheric conditions and its mixing with other components of PM within the atmosphere which can alter its optical properties (Zhao et al., 2020). Consequently, understanding the properties and interactions of BC in the atmosphere is important for both air quality and climate.

Through absorbing radiation and altering the thermal profile of the atmosphere, BC may play an important role in the enhancement of pollution episodes via the aerosol-planetary boundary layer (PBL) feedback mechanism. The mechanism can be described as follows: scattering and absorbing aerosol particles interact with solar radiation to reduce the amount of shortwave radiation (SWR) reaching the surface. The consequent reduction in SWR at the surface results in reduced buoyancy of the surface air and can impact the development of the PBL throughout the day. Specifically, aerosols suppress the development of the PBL which leads to the aerosols themselves being more concentrated at the surface and thus having stronger interactions with SWR. The effect of BC on PBL development and the aerosol-PBL mechanism is highly dependent on the properties of BC as well as the altitude of the BC layer. In theory, concentrations of BC at the surface will warm the lower layer, promoting buoyant turbulence and decreasing atmospheric stability, while a layer of BC aloft is thought to further enhance any existing temperature inversions, increasing temperatures aloft and decreasing them at the surface leading to atmospheric stabilisation (Figure 1). Upper level BC (BC above the PBL) is also believed to have a higher heating efficiency due to a combination of the lower density of air and higher incident radiation flux at higher altitudes, allowing for BC particles to absorb more radiation and heat air at a higher rate. This suggests that even low concentrations of BC at high altitudes may lead to atmospheric stabilisation. This is considered to be the so-called 'dome effect' of BC (Ding et al., 2016; Gao et al., 2015; Petäjä et al., 2016; Wang et al., 2018; Zou et al., 2017).

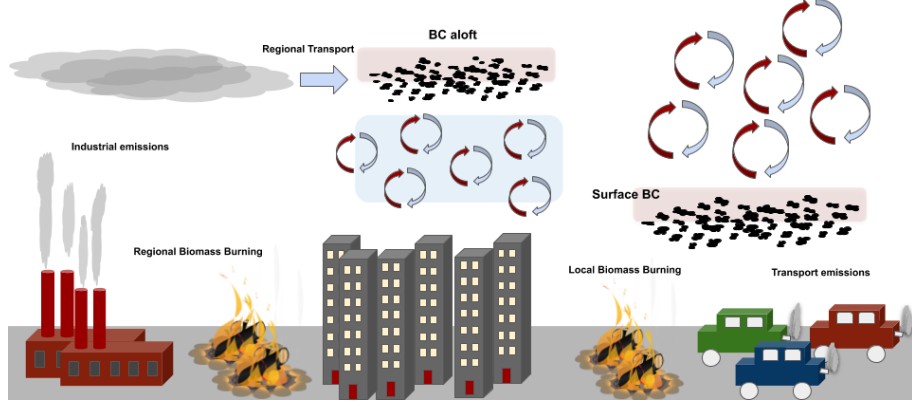

**Figure 1.** Schematic showing some of the sources of BC in Beijing, which include industrial emissions, regional and local biomass burning and emissions from transport. As well as outlining the main concepts presented in this paper of the influence of BC aloft and BC within the PBL on PBL dynamicss

The impact of BC on the aerosol-PBL feedback is dependent on several factors, including: the altitude of the BC layer, its concentration and mixing state. Ding et al. (2016) found that BC enhanced haze episodes through warming the air above the PBL and enhancing stratification of the boundary layer. Wang et al. (2018) found that surface BC promoted PBL development through warming but that this effect was negated by the stronger interactions of BC aloft which suppressed PBL growth. Ding et al. (2016) first showed the importance of the BC dome effect through conducting simulations of three megacities in Eastern China and changing the level of aerosol feedback. Through this, they directly characterised the feedback effect of BC compared to that of other aerosols. Their results showed that a maximum change in SWR due to BC occurred at the top of the PBL (around 400 m in this case) and that BC at this altitude was primarily responsible for the large suppression in PBL height, despite only making up 30 % of the column BC concentration. Furthermore, BC reduces downwelling surface SW radiation, leading to surface cooling and contributing up to 50 % of the total aerosol reduction in surface fluxes. In a 1D modelling study, Wang et. al (2018) found that BC aloft was essential in the suppression of PBL height, with surface BC increasing both turbulence and PBL height. Furthermore, they found that a BC layer close to the PBL top and internal mixing of BC with scattering aerosols (sulphate ($SO_4^{2-}$), nitrate ($NO_3^-$) and ammonium ($NH_4^+$)) significantly enhanced the dome effect of BC, leading to a reduction in PBL height of 15 %. BC is thought to exist above the PBL in Beijing due to aerosols being transported from surrounding regions by synoptic scale winds and through aerosols remaining in the residual layer overnight after being mixed through the PBL in the previous day (Wang et al., 2016; Zhao et al., 2020).

Thus far methods of examining the impact of BC on the aerosol-PBL feedback in Beijing have been with observational or regional modelling studies, with specific radiative transfer models used to calculate direct radiative effects (Ding et al., 2016, 2017; Wang et al., 2018; Zhao et al., 2020). Large eddy simulation (LES) models directly simulate turbulent motion and PBL development without the parameterisation of boundary layer processes which is necessary in regional models such as WRF-CHEM. This gives them a significant advantage in understanding and quantifying perturbations to the PBL. Previously LES

models have been used to examine the effect of absorbing aerosol layers on the development of stratocumulus and cumulus clouds. Herbert et al. (2020) examined the effect of an absorbing layer on stratocumulus clouds and thus the PBL development and rates of entrainment, finding a significant reduction in the entrainment rate the closer the absorbing layer was to cloud top. Related to dissipation of radiation fog, Maalick et al. (2016) found that in polluted conditions, BC has a warming effect close to the fog top, and BC enhances fog dissipation due to absorption of solar radiation. However, if the increase in BC concentration is accompanied with an increase in CCN and thus fog droplet concentration, the CCN effect increasing fog lifetime is much stronger than the contrasting BC effect which works to shorten fog lifetime.

In this work, we use the coupled LES-aerosol-radiation model (UCLALES-SALSA), which has previously been setup and tested in Beijing to examine the impact of BC on aerosol-PBL interactions and the implication on Beijing haze episodes (Slater et al., 2020, 2021). The high resolution of LES models and their ability to calculate turbulent fluxes and perturbations thereof, allows for isolation and quantification of the different factors impacting the 'dome effect' of BC. We use meteorological conditions from 2 days in the middle of a haze episode (2nd and 3rd Dec 2016), where a strong temperature inversion and shallow PBL already exist due to the convergence of cold northerly air masses with southerly warm air masses (Wang et al., 2019). Specifically, this work investigates the 'dome effect' of BC, through isolating the impact of BC both above and within the PBL and the impact on PBL dynamics. This paper showcases a series of idealised simulations. It is worth noting that in all simulations there is no surface heterogeneity or changes in vertical structure across the model field. Due to the lack of heterogeneity across the model field, all results presented, and plots shown are horizontal domain averages to explain the driving processes as a function of time. This paper is set out as follows. Section 2 describes model setup, including experimental setup for the different sensitivities examined, while section 3 details the results for the corresponding sensitivity studies. Section 4 briefly discusses the overall results and their implications in more detail.

## 2 Methods

### 2.1 Model Description

The model used in this study is UCLALES-SALSA, which is a large eddy simulation model (UCLALES) fully coupled to the sectional aerosol model (SALSA). UCLALES-SALSA has been used to examine the impact of aerosols on stratocumulus clouds, radiation fog, cloud seeding and to examine the aerosol-PBL feedback in Beijing (Tonttila et al., 2017; Slater et al., 2020; Tonttila et al., 2021; Slater et al., 2021). LES models resolve the three-dimensional turbulent field of wind and scalar concentrations and directly resolve most of the energy and parametrises only the smallest scale eddies. UCLALES uses the Smagorinsky-Lilly subgrid model, with leapfrog time stepping used for advection of momentum variables and forward time stepping for advection of scalar variables, based on fourth order differential equations. Boundary conditions are periodic in both horizontal directions and fixed in the vertical. The surface scheme for moisture and heat is based on a coupled soil moisture and surface temperature scheme by Ács et al. (1991), which explicitly calculates surface temperature and, sensible and latent heat fluxes. The surface scheme used in this case has been adapted and tested for the urban environment of Beijing and the

setup which includes the addition of a diurnal anthropogenic heat flux ($Q_f$) and alterations to the surface heat capacity value as detailed in Slater et al. (2020).

SALSA is a sectional aerosol model which has been fully coupled to both UCLALES and the climate model ECHAM (Kokkola et al., 2008, 2018). SALSA bins aerosol particles according to size, with three bins for aerosol particles in the nucleation mode (diameter between 3 and 50 nm) and two sets of seven parallel size bins for aerosol particles in the aitken, accumulation and course modes (diameters between 50 nm and 10 $\mu$m, which allow for aerosols to be both internally and externally mixed. In the results set out here, we assume all particles to be internally mixed. Processes including deposition of aerosols, semi-volatile condensation, nucleation and emissions are switched off but aerosol coagulation and water condensation on aerosol particles are turned on. For these simulations, organic carbon (OC), $SO_4^{2-}$, $NO_3^-$, BC and $NH_4^+$ are included. We use the same size distribution for all simulations (Table 1) and composition is varied slightly to examine the impact of fractional composition changes of BC (Table 2).

To calculate aerosol-radiation interactions, SALSA uses a four stream radiative transfer scheme based on the work by Fu and Liou (1993). This scheme is fully coupled to UCLALES to allow feedback on turbulent dynamics and is a four stream method integrating over 6 SW bands and 12 LW bands. To account for the impact of aerosol size on aerosol-radiation interactions we use pre-computed constant refractive indices, and use look up tables for the aerosol-extinction cross section, asymmetry parameter and single-scattering albedo, which are calculated as a function of the size parameter ($\alpha = \frac{D_p}{\lambda}$), where $D_p$ is the particle diameter and $\lambda$ is the wavelength of light. In this work we set all imaginary parts of the refractive indices in the SW to zero apart from that of BC which is set to values according to Bond and Bergstrom (2006). This allows us to consider BC as the only absorbing aerosol. SALSA treats internal mixing for optical properties in a simple way, through volume averaging of the complex refractive index of each component in each particle. Optical properties of the entire particle are calculated from the average refractive index of the particle according to volume as detailed in Jacobson (2005). Therefore, the potential of scattering aerosols to enhance absorption of BC through the 'lensing effect' is not considered here.

## 2.2 Experimental Setup

The work presented here is divided into three sections and specific setup for each sensitivity is detailed in the appropriate section. We perform simulations for 2nd and 3rd Dec 2016 in Beijing, varying the altitude of the aerosol layers and fractional composition. Initial meteorological conditions were taken from radiosonde profiles which are located at Beijing International Airport with measurements being taken twice per day at 8am and 8pm local time. Aerosol composition and size parameters were calculated based on ground based measurements taken at 8am on 3rd Dec 2016 during the Air Pollution and Human Health (APHH) winter field campaign (Shi et. al 2019). The initial aerosol vertical profiles were estimated based on the gradient of boundary layer profiles. The model was set up at 08:00 LST and run for 12 hours including 1 hour spin up time. The horizontal domain size was 5.4 x 5.4 km$^2$ with a resolution of 30 m and the model top was set to 1800 m with a vertical resolution of 10 m. For all simulations, the initial size distribution is outlined in Table 1. While the fractional composition of

the aerosol particles for BC and no BC simulations are detailed in Table 2. For all results, PBL top or maximum PBL height is taken as the height at which there is a maximum gradient in potential temperature ($\theta$).

| Parameter | Mode 1 | Mode 2 |
|---|---|---|
| $D_g$ (nm) | 22 | 121 |
| $\sigma_g$ | 1.28 | 1.32 |

**Table 1.** Size distribution input data - geometric mean diameter ($D_g$) and geometric standard deviation ($\sigma_g$)

| | $SO_4$ | BC | OC | $NO_3$ | $NH_4$ |
|---|---|---|---|---|---|
| **BC case** | 0.1 | 0.1 | 0.45 | 0.25 | 0.1 |
| **No BC case** | 0.2 | 0.0 | 0.45 | 0.25 | 0.1 |

**Table 2.** Volume fractional composition for BC and no BC simulations in all simulations

Individual case setup for the changing conditions examined in this paper are detailed in sections 2.2. Overall, three case study experiments with a total of 14 simulations were performed to examine the different impact of: 1) Aerosol loading at different altitudes both with and without the effect of BC (Case Aero_load), 2) Different initial meteorological conditions (Case Met) and 3) Changing concentrations of BC within the aerosol column (Case BC_load). Section 2.2.1 outlines the setup of simulations for the first case study (Case Aero_load) which examines the impact of varying the composition of aerosol layers at different altitudes to either include or not include BC. These six simulations are varied so that there are three different altitudes for an aerosol layer and each layer either has a fractional composition of 10 % BC or no BC (Table 2). In these simulations, the aerosols are only present within the specified layer, with no aerosols present initially above or below the layer. Section 2.2.2 outlines the setup for the second case study (Case Met) which focuses on examining the effect of initial meteorological conditions on the impact of BC heating within the PBL on boundary layer. For these four simulations only a surface aerosol layer is considered and the initial meteorological conditions are either taken from the morning of 02 Dec or 03 Dec 2016. Section 2.2.3 describes the setup for the third case study (Case BC_load) simulations which examines the impact of varying the fraction of BC in different vertical layers for simulations where aerosols are present throughout the column.

### 2.2.1 Case Aero_load

To isolate the impact of BC and the altitude of the BC layer on the aerosol-PBL feedback, we performed sensitivity studies with meteorological conditions taken from measurements on the morning of 3rd Dec 2016. In the simulations presented in this section, we varied the altitude of the aerosol layer as well as the fractional composition of the aerosols, to either have BC or no BC within the layer (Table 2). We included aerosol layers with identical mass mixing ratios within the PBL (0-350 m), at and above PBL top (500-950 m) and high aloft (700-1150 m) as shown in Fig. 2 (red, blue and cyan lines respectively).

Maximum PBL height (top) was considered to be 510 m based on simulations performed on this day without aerosols. For all simulations in this section, total aerosol loading was kept constant, while the composition was varied as detailed in Table 2 and concentrations and altitudes of each of the aerosol layers as shown in Fig. 2.

| Case name | Description |
|---|---|
| **Aero_load_BCsurf** | Aerosols (including BC) are only included initially in a layer between 0-350 m |
| **Aero_load_noBCsurf** | Aerosols (not including BC) are only included initially in a layer from 0-350 m |
| **Aero_load_BC500** | Aerosols (including BC) are only included initially in a layer from 500-950 m |
| **Aero_load_noBC500** | Aerosols (not including BC) are only included initially in a layer from 500-950 m |
| **Aero_load_BC700** | Aerosols (including BC) are only included initially in a layer from 700-1150 m |
| **Aero_load_noBC700** | Aerosols (including BC) are only included initially in a layer from 700-1150 m |

**Table 3.** Table detailing the outline of each of the six simulations for the first case study (Case Aero_load).

Table 3 details the names and brief description of each of the simulations in this case. Figure 2 shows the vertical aerosol profiles for each of the simulations. Fig. 2a shows the variation in BC concentrations for the 3 simulations which include BC (solid lines), while the simulations without BC (dashed lines) have no BC through the whole layer. Fig. 2b shows the variation in $SO_4^{2-}$ aerosol concentrations. As shown in Table 2 and Fig. 2c, for simulations without BC, the initial aerosol concentrations are the same as the simulations with BC but the fractional composition is varied so that there is double the concentration of $SO_4^{2-}$. Fig. 2c shows the overall vertical variation in aerosol layers for the different simulations, where the BC and no BC simulations for each of the 3 different layers have the same aerosol vertical profiles.

### 2.2.2 Case Met

To examine how sensitive BC heating at the surface is to the initial meteorological conditions, particularly the strength of temperature inversions both in the morning and throughout the day, we included an aerosol layer with BC near the surface for simulations on 2nd Dec and compared the results to simulations performed on 3rd Dec. We then compared each simulation to a reference simulation, hereafter referred to as the base case which did not include aerosol-radiation interactions. The difference in initial meteorological conditions are outlined in Fig. 3. On the 2nd Dec, relative humidity is lower (Fig. 3b), while surface wind speeds are higher (Fig. 3c) and the total temperature inversion throughout the profile is weaker (Fig. 3a) than on 3rd Dec. Furthermore, there is a shallow layer with a sharp temperature inversion at the surface in the 2nd Dec initial profile. However, the free tropospheric lapse rates (above 1000 m) for 3rd Dec are higher than on 2nd Dec, indicating a higher degree of stability on this day. The aim of this case study was to examine the influence of these initial conditions on BC heating within the PBL and the associated impact on PBL dynamics.

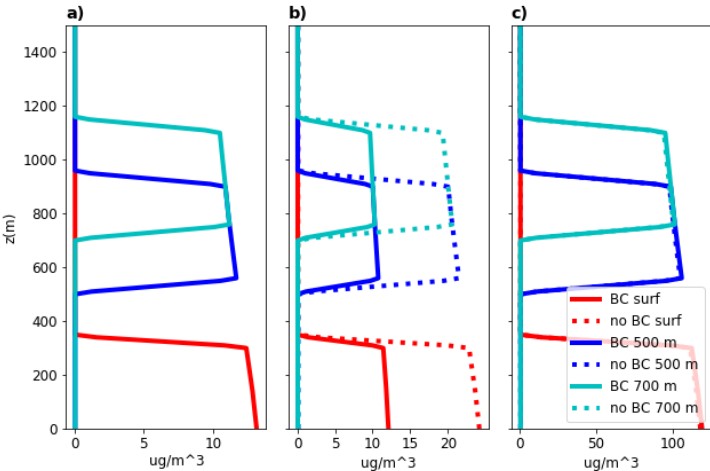

**Figure 2.** Initial mass concentrations of a) BC, b) $SO_4^{2-}$ and c) Total aerosol concentrations for simulations in case 1. Different heights were chosen for the initial aerosol layers, which were 0-350 m (red lines), 500-950 m (blue lines) and 700-1150 m (cyan lines). Simulations which included BC are depicted by solid lines and those without are dashed lines

| Case name | Description |
|---|---|
| **Met_0212_noaero** | Initial meteorological conditions taken from the morning of 02/12/2016 - no aerosols included in the simulation |
| **Met_0212_aero** | Initial meteorological conditions taken from the morning of 02/12/2016 - aerosols including BC are only included below 400 m |
| **Met_0312_noaero** | Initial meteorological conditions taken from the morning of 02/12/2016 - no aerosols included in the simulation |
| **Met_0312_aero** | Initial meteorological conditions taken from the morning of 03/12/2016 - aerosols including BC are only included below 400 m |

**Table 4.** Table detailing the outline of each of the four simulations for the second case study focusing on the impact of initial meteorological conditions (Case Met).

For these simulations, the aerosol profiles for each day were kept the same so the only variable was the different initial meteorological conditions. The included aerosol profiles are the same as for case Aero_load_$BCsurf$ in section 2.2.1 (solid red lines in Fig. 2).

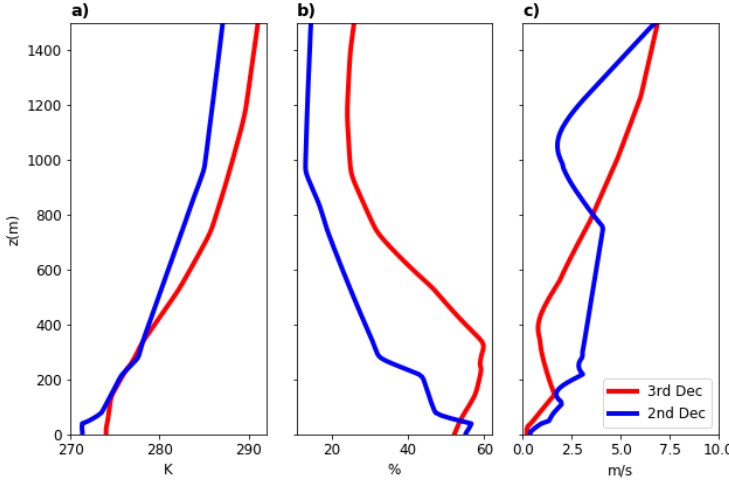

**Figure 3.** a) Potential temperature, b) Relative Humidity and c) Wind Speed profiles at 9am for 3rd Dec (blue) and 2nd Dec (red)

### 2.2.3 Case BC_load

Simulations in this setup examined the impact of changing the fractional composition of BC in different aerosol layers. The total aerosol loading in the column was kept constant in each simulation but the composition in different layers was varied as is detailed for BC and No BC cases in Table 2. This setup was done as a proxy to examine the idea of reducing BC at the surface without tackling regional emissions of BC which may get transported from surrounding areas leading to high concentrations above the PBL. Table 5 outlines a brief description of variations used in each simulation for the BC_load case.

| Case name | Description |
|---|---|
| BC_load_noBC | No BC included at all altitudes |
| BC_load_500 | BC included above 500 m only |
| BC_load_1000 | BC included above 1000 m only |
| BC_load_full | BC included at all altitudes in the profile |

**Table 5.** Table detailing the outline of each of the four simulations for the third case study focusing on the impact of changing BC loading in different aerosol layers (Case BC_load).

The setup here uses meteorological conditions from 3rd Dec 2016 as outlined in section 2.2.1 but includes aerosols throughout the column, rather than just in specific layers. Details of the variation in BC concentrations are outlined in Figure 4, and include BC throughout the column (red lines), BC above 500 m (blue lines), BC above 1000 m (cyan lines) and no BC (green lines). Each of the individual simulations and the corresponding names of each of the simulations are detailed in Table 5. As in the setup described in section 2.2.1, the aerosol loading throughout the column is the same with changes in composition for

the 'BC' and 'No BC' cases as detailed in Table 1. Fig. 4a shows the variation in BC loading, Fig. 4b shows the variation in

$SO_4^{2-}$ loading and Fig. 4c shows total aerosol vertical profiles for all simulations.

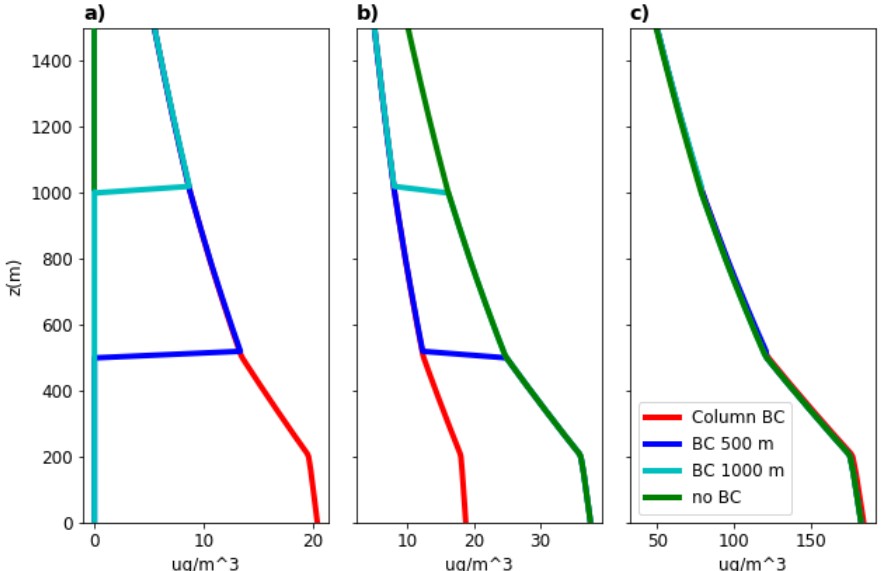

**Figure 4.** Initial mass concentrations of a) BC, b) $SO_4^{2-}$ and c) Total aerosol for each of the experiments outlined. For simulations with the same aerosol concentration but varying composition across the column to have: BC through the column (red), BC above 500m (blue), BC above 100m (cyan) and no BC throughout the column (green)

## 3   Results

The results here are separated into 3 sections based on the experimental setup outlined in section 2.2.1-2.2.3. Section 3.1 outlines results from Case Aero_load simulations performed as outlined in section 2.2.1 for vertically varied aerosol layers, section 3.2 shows results from Case Met simulations looking at the effect of initial meteorological conditions as described in the setup in section 2.2.2. Finally, section 3.3 showcases results from case BC_load simulations, which examines the effect of varying BC loading in different vertical layers as described in the setup in section 2.2.3.

### 3.1   Case Aero_load - Vertically varied aerosol layers

Results from varying the height of aerosol layers in the column, with and without BC aerosols show that at all altitudes, BC in the aerosol layer has more impact on PBL dynamics than the effect of scattering aerosols alone (No BC simulations) (Table 6). Specifically, simulations without BC have a small effect on surface temperature, sensible heat flux and PBL height. At the surface (Aero_load_BCsurf), BC decreases downwelling surface SWR by 4 % compared to including only scattering aerosols (Aero_load_noBCsurf), due to the increased absorption caused by increasing the BC fraction. This decreases maximum surface sensible heat flux (SHF) by 11 % and slightly decreases surface temperature by 0.02 %. . However, BC absorption of SWR at this altitude causes warming of the air layer above the surface (Aero_load_BCsurf), leading to a slight increase in air temperature at 10 m (0.07 %) and PBL height (0.4 %) compared to the Aero_load_noBCsurf simulation (Table 6).

For further analysis, we calculated SW heating rate at time (t) by BC as the change in SW radiative flux ($\downarrow$SW - $\uparrow$SW),divided by specific heat capacity of air ($C_p$) multiplied by density of air ($\rho$) as in equation 1. Where each timestep (t+$\Delta$t) here is 2 minutes.

$$SW\,Heating\,Rate_{(t)} = \frac{(\downarrow SW - \uparrow SW)_{(t+\Delta t)} - (\downarrow SW - \uparrow SW)_{(t)}}{C_p * \rho} \tag{1}$$

We calculate that the heating rate of BC varied between 0.1-0.2 K/h which could lead to a maximum heating of the PBL throughout the day in wintertime Beijing of 1.6 - 2 K. If the temperature inversion during the day was small (1-3 K), this additional heating by BC within the PBL and at PBL top could break the temperature inversion at PBL top and increase PBL height. However, under a strong temperature inversion as on 03 Dec, this heating within the PBL was not strong enough to reduce the temperature inversion fully and so there was a very small increase in PBL height. Consequently, BC heating within the PBL in this case only resulted in a very small (0.4 %) increase in PBL height (Table 6). The strength of the temperature inversion and the growth of the PBL throughout the day are impacted by the synoptic scale meteorological conditions. Furthermore, the low albedo (0.2) and high heat capacity of the underlying surface, typical of an urban environment, mean that BC at the surface will have a lower impact than studies which have examined the effect of polluted environments over high albedo surfaces, for example, clouds or rural environments (Wang et al., 2018).

Figure 5 outlines the changes in SW downwelling and upwelling radiation as well as the SW heating rate due to the presence of BC in the aerosol layers for all three simulations which include BC (Aero_load_BC700, Aero_load_BC500,

| | SHF (W/m$^2$) | PBL Height (m) | Surface T (K) | ↓ SWR surface (W/m$^2$) | ↑ SWR top (W/m$^2$) | T at 10 m (K) |
|---|---|---|---|---|---|---|
| **Aero_load_BCsurf** | 98.19 | 511.43 | 285.04 | 490.18 | 102.35 | 279.55 |
| **Aero_load_noBCsurf** | 111.7 | 509.43 | 285.11 | 511.02 | 108.41 | 279.31 |
| **Aero_load_BC700** | 105.18 | 494.99 | 284.69 | 491.72 | 101.91 | 279.02 |
| **Aero_load_noBC700** | 111.85 | 510.19 | 285.11 | 515.56 | 109.44 | 279.30 |
| **Aero_load_BC500** | 105.0 | 474.28 | 284.69 | 494.00 | 102.39 | 279.04 |
| **Aero_load_noBC500** | 111.84 | 508.57 | 285.11 | 515.52 | 109.44 | 279.30 |
| **Base case** | 112.29 | 510.10 | 285.12 | 516.14 | 109.28 | 279.31 |

**Table 6.** Maximum sensible heat flux (SHF), planetary boundary layer (PBL) height (taken as the height with the largest gradient in $\theta$), surface temperature (T), downwelling shortwave radiation (↓SWR) at the surface, upwelling shortwave radiation (↑SWR) at model top (1800 m) and air temperature (T) at 10 m. Values are the maximum between 12:00 and 16:00 local standard time (LST)

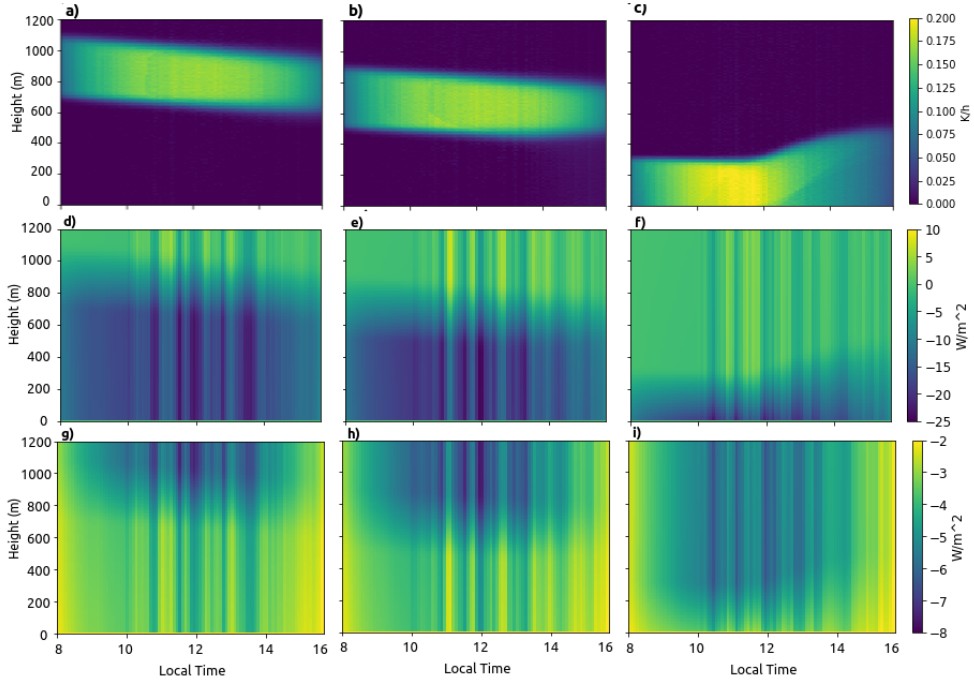

**Figure 5.** SW heating rate (a-c), loss in downwelling (d-f) and upwelling (g-i) SW radiation due to BC inclusion in the aerosol layer. All plots are normalised against no BC simulations (BC in the aerosol layer - no BC in the aerosol layer) for Aero_load_BC700 (a,d and g), Aero_load_BC500 (b, e and h) and Aero_load_BCsurf (c, f and i) simulations

Aero_load_BCsurf from left to right respectively). In the case of BC aloft (Aero_load_BC700) there is a reduction in downwelling SW radiation both within and under the aerosol layer (Fig. 5 d) and a reduction in SW upwelling radiation (Fig. 5 g)

through the layer due to absorption. The reduction in downwelling radiation beneath the aerosol layer leads to a cooling effect

due to less available radiation heating the air. Meanwhile the absorption of SW radiation in the aerosol layer causes heating. This results in a change in the thermal profile of the atmosphere and enhances the temperature inversion, decreases maximum PBL height and atmospheric temperature (Table 6).

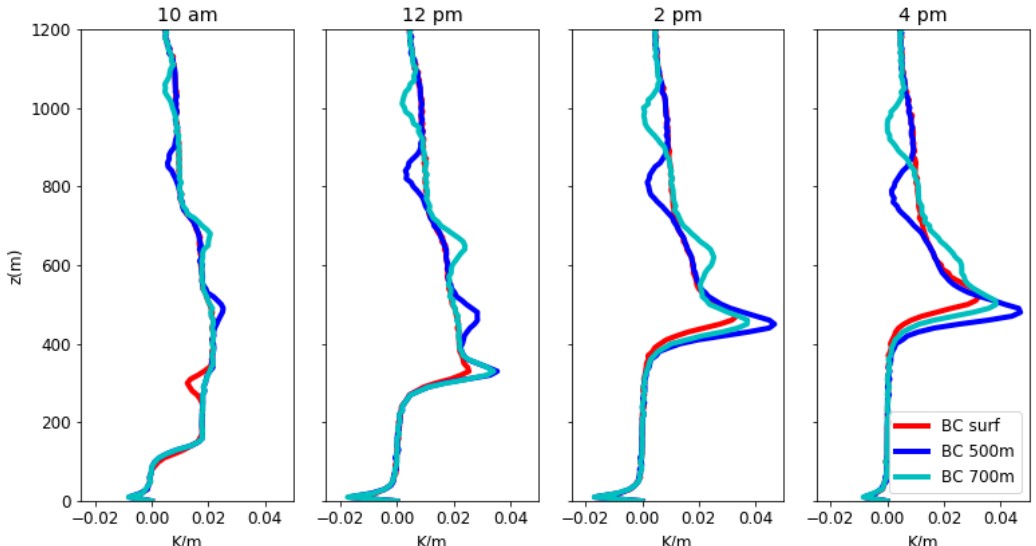

**Figure 6.** Potential temperature lapse rate for simulations on 3rd Dec at 10am, 12pm, 2pm and 4pm for Aero_load_BCsurf (red), Aero_load_BC500 (blue) and Aero_load_BC700 (cyan) simulations

Aerosol layers with identical mass mixing ratios are initialised between 500-950 m and 700-1150 m to examine the impact of the altitude of the aerosol layer for aerosols above the PBL. The higher aerosol layer (700-1150 m) has less of an impact on

PBL height than simulations with the lower aerosol layer (500-950 m). For example, Table 6 shows that case Aero_load_BC500 reduces maximum PBL height by 6.7 % compared to the base case whereas case Aero_load_BC700 only reduces maximum PBL height by 2.96 %. When there are aerosols at 500m, aerosols can become entrained into the upper PBL, as the PBL develops. This results in a strong heating at the top of and above the PBL, causing a decrease the larger decrease in PBL height compared to when the aerosol layer exists higher aloft. Figure 6 shows the potential temperature lapse rate throughout the day

for each of the aerosol layers. This shows the inversion for including the aerosol layer at 500 m is much stronger than for the 700 m aerosol layer. This causes the larger suppression of PBL development observed under these conditions.

### 3.2 Case Met-Varied Initial Meteorological Conditions

In these simulations we examined the sensitivity of BC surface heating on turbulent dynamics to different initial meteorological conditions. Specifically, we assessed whether BC at the surface can cause heating to a large enough extent to overcome the temperature inversion and enhance PBL development, under different initial conditions. On the 2nd Dec, simulations with no aerosols (Met_0212_noaero) show a temperature inversion above PBL top (400-800 m) of ∼ 4 K at 14:00 LST, compared to ∼ 7 K on 3rd Dec (500-900 m) (Figure 7). Consequently, including BC at the surface shows a larger enhancement in turbulence for 2nd Dec, when the initial temperature inversion is lower. In this case, BC at the surface causes a 5% increase in PBL height, increasing TKE and minimising the decrease in sensible heat flux. This is despite the change in SW downwelling and upwelling radiation being similar for both days (Figure 8).

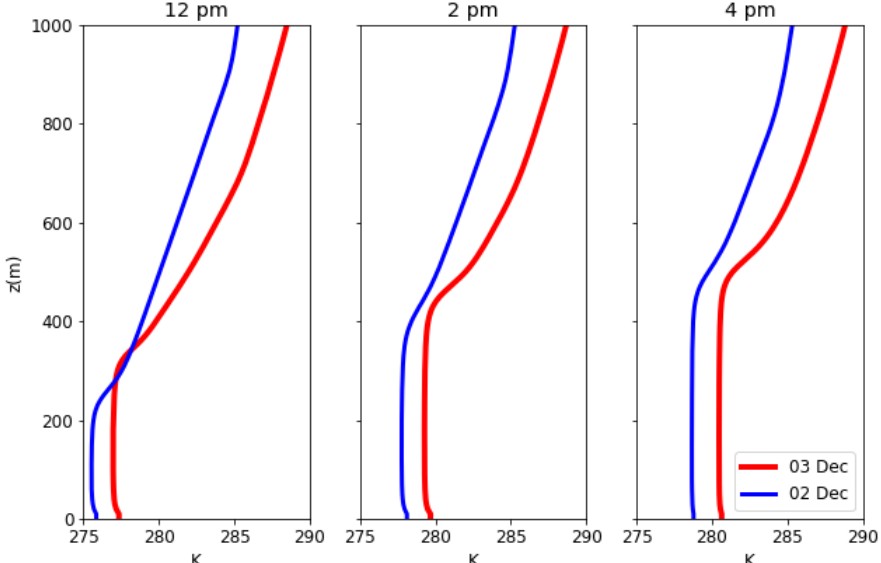

**Figure 7.** Potential temperature profiles for Met_0312_noaero simulations (red lines) and Met_0212_noaero (blue lines) at 12pm, 2pm and 4pm LST

Due to the variation in initial meteorological conditions such as humidity, temperature and wind (Figure 3), it is difficult to ascertain whether the difference in PBL development caused by BC surface heating is a direct impact of the strength of the temperature inversion. However, it is clear that at least for the case of BC within the PBL, meteorological conditions affect the magnitude of surface heating on PBL development. These results highlight the susceptibility of the aerosol-PBL feedback to initial conditions as has been outlined in previous work by Slater et al. (2020) and Slater et al. (2021). In both cases, the BC concentrations cause a heating rate of 0.2 K/h at the surface, which leads to increases in the surface air temperature. Although there is increased turbulence caused by BC heating on 2nd Dec, the level of heating within and at the top of the PBL is not enough to overcome the strong temperature inversion and the stagnation caused by aerosol-PBL feedback. Consequently,

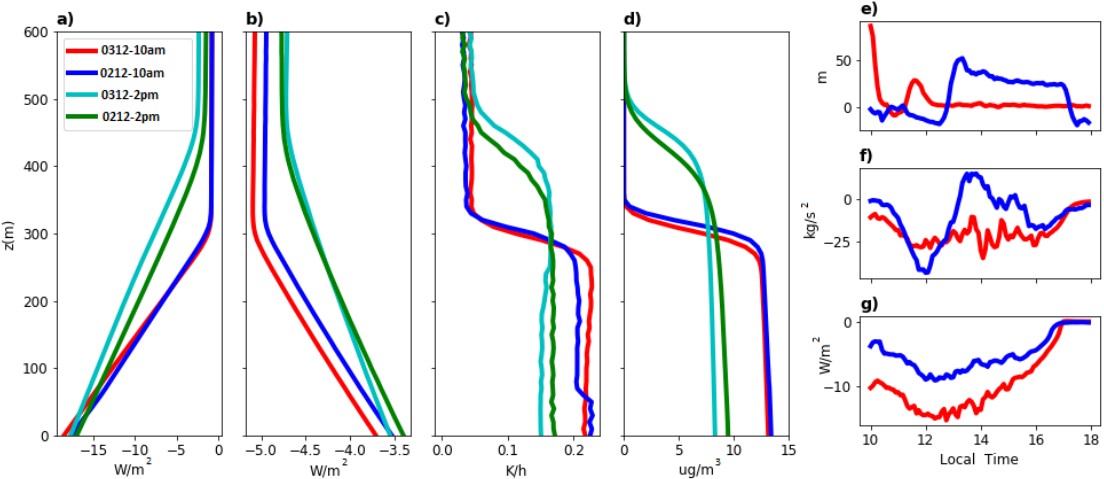

**Figure 8.** Change (surface aerosol - no aerosols) in a) downwelling and b) upwelling SW radiation, c) SW heating rate, and d) BC concentrations for simulations on 3rd Dec (red and cyan) and 2nd Dec (blue and green) at 10am (red and blue) and 2pm (cyan and green). (e-g) Change (surface aerosol - no aerosols) in e) PBL height, f) Vertical integral of Turbulent Kinetic Energy (TKE) and g) Sensible Heat Flux for 3rd Dec (red) and 2nd Dec (blue)

BC heating within the PBL under these conditions will be unlikely to promote haze dissipation due to the strength of the temperature inversion.

Figure 9 shows the potential temperature lapse rate at 12pm, 2pm and 4pm for simulations with no aerosols and with BC at the surface for 2nd Dec and 3rd Dec. This shows BC at the surface reduces the temperature inversion at PBL top in both cases. Furthermore, at 2pm the PBL top is higher on 2nd Dec for simulations including BC at the surface (Met_0212_aero) Here, compared to 3rd Dec (Met_0312_aero), the heating within the PBL and at PBL top appears to be almost strong enough to break the temperature inversion at PBL top and enhance PBL development. As can be seen in Fig. 8d, the aerosol layer becomes vertically mixed throughout the day as the PBL develops. These results show that under conditions with a weaker temperature inversion, BC heating at the surface enhances turbulent mixing to increase PBL development. If the heating caused by BC at the surface is strong enough or under conditions where the temperature inversion is weaker, this may allow for BC to become vertically mixed to high levels. When the PBL collapses overnight, this leads to BC being present above the PBL which may enhance atmospheric stagnation and enhance the intensity of pollution events.

### 3.3 Case BC_load- Vertically varied BC layers

Section 3.1 shows the aerosol radiative forcing and perturbations due to BC are higher than the scattering effect of other aerosols. However, the case Aero_load only identifies the effect of either aerosol concentrations within or above the PBL, where they can exist both within and above the PBL for several reasons. Here, we examine the idea of fully reducing BC at the surface as a proxy to decreasing BC emissions locally, where other species are still present. So for example targeting sources

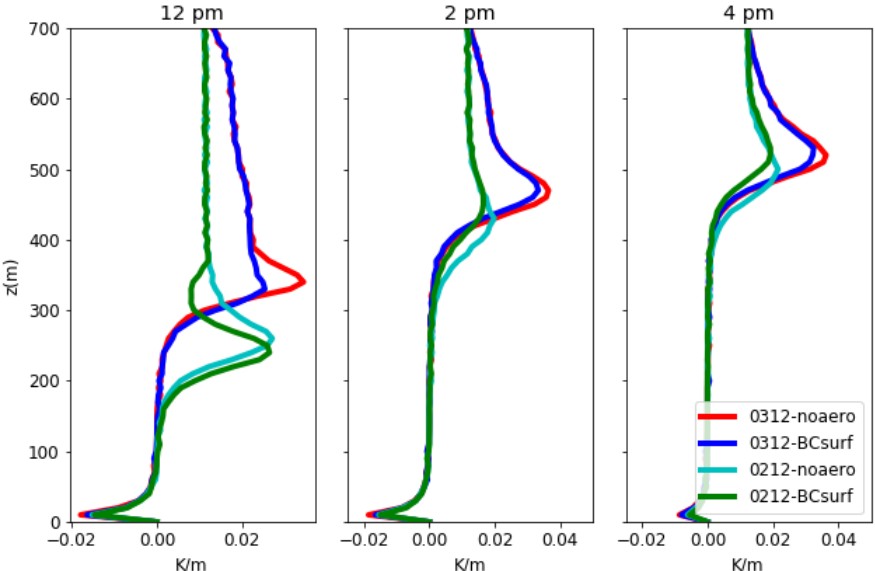

**Figure 9.** Potential Temperature Lapse Rate at 12pm, 2pm and 4pm for simulations with surface aerosols (blue and green) and no aerosols (red and cyan) on 2nd Dec (cyan and green) and 3rd Dec (red and blue)

of BC, such as biomass burning without tackling other sources of inorganic aerosols or volatile gases. BC aloft is considered to be brought into Beijing through regional transport or entrainment from a polluted residual layer. A study by Ferrero et al. (2014) suggested that the impact of local BC emissions will heat the PBL and lead to pollutant dissipation through promoting atmospheric buoyant turbulence. Results from section 3.1 and 3.2 show the reasonably low impact of BC at the surface in
enhancing PBL development, compared to the suppression caused by the BC layer at PBL top. Furthermore, if BC from the surface gets mixed into the residual layer it will negatively impact turbulent mixing the next day. This section looks at including BC and other aerosols both within and above the PBL and changing the relative BC concentration in the column.

In this section, we include aerosols throughout the column and varied the fractional aerosol composition to have BC (BC_load_full) and no BC (BC_load_noBC) throughout the profile and BC above 500m (BC_load_500) and 1000 m (BC_load_1000)
(Figure 4). Our results show that including BC both within and above the PBL causes a large reduction in PBL height (17 %) compared to no BC (Table 7). In section 3.1 and 3.2 simulations with BC have a slightly higher PBL height compared to those without (Table 3). Therefore, the decrease in PBL height for these simulations (BC within and above the PBL) indicates that the potential enhancement in turbulence by BC within the PBL (as seen in sections 3.1 and 3.2) is eclipsed by the effect of BC above the PBL which acts strongly to prevent PBL development through the day. This is likely due to the low level of SWR
available for BC heating at the surface in the full column BC simulations, due to absorption by BC at higher altitudes.

Table 7 shows that including BC has a significant impact on reducing SW downwelling and upwelling radiation, which consequently feeds back and reduces surface temperature and sensible heat flux. Simulations including BC across the entire

| | SHF (W/m$^2$) | PBL Height (m) | Surface T (K) | ↓ SWR surface (W/m$^2$) | ↑ SWR top (W/m$^2$) | T at 10 m (K) |
|---|---|---|---|---|---|---|
| **BC_load_1000** | 98.40 | 482.84 | 284.43 | 479.90 | 98.42 | 278.85 |
| **BC_load_500** | 98.42 | 423.51 | 283.78 | 455.60 | 91.22 | 278.85 |
| **BC_load_full** | 84.78 | 419.52 | 283.85 | 432.00 | 84.77 | 279.06 |
| **BC_load_noBC** | 111.03 | 507.39 | 285.10 | 503.79 | 107.90 | 279.29 |

**Table 7.** Maximum sensible heat flux (SHF), planetary boundary layer (PBL) height (taken as the height with the largest gradient in $\theta$), surface temperature (T), downwelling shortwave radiation (↓SWR) at the surface, upwelling shortwave radiation (↑SWR) at model top (1800 m) and air temperature (T) at 10 m. For including aerosols throughout the column but changing the composition for BC_load_1000, BC_load_500, BC_load_full, and BC_load_noBC simulations. Values are the maximum between 12:00 and 16:00 local standard time (LST)

column (BC_load_full), show the largest decrease in downwelling and upwelling SW radiation, due to the overall larger columnar concentration of BC. Including BC at the surface (BC_load_full)) leads to higher air temperature at 10 m compared to simulations with BC aloft (BC_load_500 and BC_load_1000) only, but lower air temperature at 10 m than not including BC at the surface (BC_load_noBC). This is likely due to BC throughout the column absorbing radiation, which leads to heating but also reduces the amount of SW radiation reaching the air at the surface, consequently reducing surface air temperature. This work shows that any increase in PBL height due to BC at the surface is outweighed compared to the stronger impact of BC above and at PBL top (BC_load_500), which results in the largest decrease in PBL height for simulation with BC across the column (BC_load_full).

Figure 10 shows the BC heating rate per unit mass of BC, taken as the heating rate for BC_load_full - BC_load_noBC simulations. This shows the heating rate per unit mass of BC increases with height as suggested by Wang et al. (2018). Firstly, we see a strong heating effect increasing up to the bottom of the PBL with a decrease in heating rate across the PBL, and a further constant increase above the PBL. The larger heating rate of BC at higher altitudes is thought to be due to the higher incident radiation available for BC absorption. Consequently, the heating caused by BC in the atmosphere and the effect on PBL development will be dependent on the altitude of the BC layer as well as the total BC within the aerosol column. This may be important when examining the impact of BC within the PBL as if BC also exists aloft, as there will be less SW radiation reaching the surface due to BC at higher altitudes and consequently as shown here, BC heating in the lower layers will be smaller.

## 4  Discussion

The results here show that BC causes heating in the atmosphere, and that absorption of solar radiation by BC has a larger impact on the temperature profile of the PBL compared to the effect of scattering aerosols. Specifically, BC can cause surface cooling through reducing SW radiation reaching the surface. In this study, BC causes heating in the aerosol layer at a rate of around 0.01-0.016 K h$^{-1}$/$\mu$gm$^{-3}$ of BC (Figure 10) , which is similar to that proposed by Ding et. al (2016) and Wang et. al

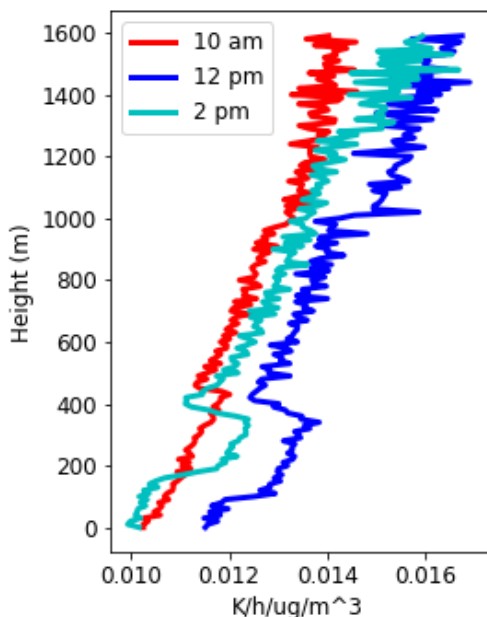

**Figure 10.** SW heating rate per unit mass of BC at 10am, 12pm and 2pm on 3rd Dec for simulations with BC throughout the column

(2018). For the concentrations used in this study, this leads to an overall heating of around 0.2 K/h or 1-1.6 K/day, showing the potential impacts of BC on climate through warming the atmosphere. Furthermore, this work directly investigates the impact of BC altitude on the layer, where BC is considered to be the only absorbing aerosol.

From examining the potential temperature lapse rate, we can see that BC at 500 m has a larger impact on reducing the temperature inversion at PBL top compared to simulations with BC at 700 m. This suggests that the absorbing layer has 325 the most impact when it exists at PBL top (500 m in these case studies). Furthermore, throughout the day the aerosol layer becomes entrained into the PBL as it develops. When aerosols are included throughout the column, we observe an enhanced effect on PBL suppression, with a decrease of 16 % compared to not including BC and only a slightly lower impact than having BC throughout the entire column (Section 3.3). When there are aerosols throughout the column, BC at the surface will receive less SW downwelling radiation compared to BC aloft due to the interactions of the aerosols above it preventing SW 330 downwelling reaching lower levels. Consequently, there will be more SW radiation available for the BC aloft to absorb and heat the atmosphere. Figure 10 shows that the heating rate per unit mass of BC is higher aloft than at the surface, meaning that the impact of PBL suppression by BC aloft will often negate the impact of surface BC promoting PBL development.

At the surface, including BC increases air temperature, but decreases sensible heat flux through reducing the amount of SW radiation reaching the surface. We examine the impact of BC at the surface for a case study including the 2nd and 3rd Dec 335 2016 and find that the magnitude of the impact is different on each day. This highlights the potential for initial meteorological conditions to influence the effect of BC on the aerosol-PBL feedback effect. In section 3.1 and 3.2, we identify that the strong

temperature inversion at PBL top throughout the day on the 3rd Dec prevents BC heating from enhancing PBL development. The impact of BC within the PBL on PBL height in our simulation is small (0.26 % increase), compared to a 4-6 % increase suggested by Wang et al. (2018), despite the heating rates due to BC being similar. We believe this is due to the strength of

340 the temperature inversion caused by the initial conditions on 3rd Dec (Figure 3a). A temperature inversion, often known as the capping inversion is the sharp increase in temperature at PBL top and can be enhanced or diminished by aerosol-radiation interactions (Stull, 2015). Warming of the air at PBL top by BC can reduce the difference in temperature between the PBL and free atmosphere, reducing the temperature inversion and making it easier for air parcels to move upwards. In some cases, this will increase PBL height. In the work by Wang et al. (2018), the temperature inversion at PBL top to 400m above PBL top at

345 14:00 LST without aerosol inclusion is $\sim$ 3 K, in our case it is $\sim$ 7 K.

Temperature inversions and stable conditions are frequently brought about by synoptic condition changes in wintertime Beijing, are strengthened over the pollution episode as aerosols cool the surface. Wang et al. (2019) examined the causes of the pollution episode examined in this work (01-04 Dec 2016) to understand the influences of synoptic scale meteorology and aerosol-PBL feedback effect on the pollution episode. They suggest the strong temperature inversion on the 3rd Dec is due to

350 both the impact of synoptic conditions and the aerosol-PBL feedback from the previous day causing surface cooling. Overall, we find that surface BC causes warming and enhances turbulence. This increases PBL height by 0.26 % on 3rd Dec due to the strong initial temperature inversion (7.0 K in the lowest 500m at 10am) but increases PBL height by 5 % on 2nd Dec due to the weaker temperature inversion (4.2 K in the lowest 500 m at 10am). However in these conditions, the heating rate is still not enough to fully weaken the strong temperature inversion (Figure 9).

A recent study by Ma et al. (2020) utilises an LES model to investigate the impacts of the altitude of absorbing and scattering aerosol layers on planetary boundary layer height. Although, in their work they do not specifically treat the aerosol population but focus on changing the aerosol optical depth and aerosol optical properties directly. Their work agrees with the work published here, to show that absorbing aerosols above the PBL will suppress turbulence and PBL growth due to the so-called "the dome effect". Their results also show an increase in PBL height due to absorbing aerosols at the surface promoting turbulent

motion, which they term "the stove effect". In contrast to the work presented here, their work shows a much greater increase in PBL height due to absorbing aerosols within the PBL, however, the $PM_{2.5}$ concentrations over the period investigated by Ma et al. are much lower than in this work. Similarly, the maximum PBL height in that work (around 1000 m) is much higher than in our work. Consequently, it is likely that the specific meteorological conditions are different in Ma et al. and this study and that, as highlighted in section 3.2 of this paper, this has a significant impact on the ability of absorbing aerosols to increase

PBL height or the magnitude of "the stove effect".

This work shows that BC heating may significantly increase the PBL height, specifically under conditions with weaker capping inversions. However, surface BC has only a small effect on enhancing PBL development under strongly stagnant meteorological conditions. It is therefore unlikely that under polluted conditions in wintertime Beijing BC heating at the surface will strongly impact the aerosol-PBL feedback loop to lessen the severity of pollution episodes. It may be however,

that under certain conditions when the temperature inversion is low that the heating of BC impacts PBL development and

may enhance the recovery phase of a pollution event though we have not focused on this aspect in this paper. We calculate a heating rate of around 0.2 K/h, and assuming a daytime heating in Beijing winter of 8 hours, the heating at PBL top could reach around 1.6 K which is too small to change temperature inversions which are common during haze episodes and are typically 4-7 K. However, in cases with weaker temperature inversions (1-2 K), the heating caused by surface BC may be strong enough to cause modifications to the PBL development. This allows for increased vertical mixing of aerosols and may allow for BC to be mixed higher into the PBL, when cooling of the surface overnight results in a very stable shallow nocturnal boundary layer forming which could leave some of the polluted aerosol layer aloft. The next day this polluted absorbing layer will heat the layer above the PBL, thus changing the temperature profile of the PBL to reduce buoyancy. This reduces PBL height and enhances the aerosol-PBL feedback to increase surface $PM_{2.5}$ and intensify pollution episodes (Figure 11). Our results show that BC above the PBL has more impact than BC below and consequently if BC is present throughout the column, the effect of suppressing turbulent motion by BC is greater than the enhancement effect.

In performing simulations including BC throughout the column (case BC_load) this work can directly show that the impact of BC heating within the PBL is negated by the stronger impact of BC aloft, which absorbs a significant proportion of SW solar radiation. This means ther is less absorption of SW radiation by BC within the PBL or at lower altitudes. This work therefore adds on to the studies by Ding et al. (2016), which only shows the effect of BC at and above PBL top, and Wang et al. (2018) which examines the impact of BC layers at different altitudes separately rather than the effect of multiple BC layers. Our work also shows the importance of initial conditions on the BC surface heating effect as outlined for aerosols in general by Slater et al. (2020) and Slater et al. (2021). This is important as these conditions change over the course of the haze episode, with PBL height found to decrease by as much as 50 % due to synoptic influences alone (Wang et al., 2019; Slater et al., 2021). In the work by Wang et al. (2018) only one set of meteorological conditions are examined, which limits the applicability of the results to periods with similar conditions. While our work shows that conditions on 02 Dec lead to a PBL enhancement of 5 %, compared to 0.4 % on 03 Dec.

## 4.1 Limitations and novelty of this study

Using a coupled aerosol-radiation LES model in this study allowed for direct investigation and quantification of the impact of absorbing aerosols on boundary layer dynamics. There is an array of benefits for using a high resolution model which directly calculates rather than parameterises turbulent fluxes, for the investigation of heavy pollution episodes. As previously highlighted in this paper, the importance of aerosol-boundary layer feedbacks on heavy pollution episodes, particularly in the megacity of Beijing has been made clear in the literature over the last decade. Primarily, these studies have utilised measurements of aerosol properties alongside measurements of boundary layer dynamics and meteorology to infer the impact and relationship between aerosols and boundary layer dynamics. Modelling studies of the aerosol-boundary layer feedback mechanism have mostly utilised regional models such as WRF-CHEM which do not directly resolve turbulent flows. Therefore, the work presented here showcases a novel methodology for investigation of the contrasting impact of absorbing aerosols

on boundary layer dynamics and the usefulness of employing such high resolution eddy resolving coupled aerosol-dynamic models to examine physical processes and interactions which can severely influence pollution episodes.

However, applying the idealised model setup to a polluted megacity has limitations. Not being able to fully account for changes in synoptic conditions or understand the impact of regional transport of aerosols on the pollution episodes is a major limitation of the work presented here. Similarly, this work does not look at the impact of secondary aerosol formation which has been found to be a major factor in the rapid increase in $PM_{2.5}$ concentrations during Beijing haze episodes. Specifically, with regards to the effect of black carbon on aerosol-boundary layer feedback, a current limitation of the work presented here

is that it doesn't account for the absorption enhancement of black carbon by scattering aerosols through the lensing effect (Liu et al., 2017). Furthermore, in this work we consider the only absorbing aerosol to be black carbon, while brown carbon (BrC) has been found to be an important absorber of radiation in several polluted megacities. Although, BrC isn't as strong an absorber of radiation as black carbon, its presence in high concentrations in polluted urban environments means its impact on these feedbacks should not be discounted (Cheng et al., 2016; Xie et al., 2019). There is scope within UCLALES-SALSA to

change the refractive indices and mixing type to reflect some features related to changes in absorption. However, in this work, for simplicity and to allow for the ability to isolate different effects these were not considered.

## 4.2    A mechanism for the impact of BC on Beijing pollution episodes

Combining all the results presented in this paper as well as other research by Wang et al. (2018) and Ding et al. (2016), here we detail a potential mechanism for the influence of BC on air pollution episodes in Beijing (Figure 11). Although this mechanism

has not been fully tested in this work due to computational cost, we hypothesise that locally emitted BC which heats the PBL could promote PBL development (section 3.2), resulting in the BC becoming well mixed through the PBL. When the nocturnal boundary layer forms, the BC will remain in the residual layer overnight and exist above the PBL the next day. This would then suppress PBL development as shown in section 3.1 and 3.3 of this work and in the work by Ding et al. (2016). However, if synoptic conditions on the next day changed to weaken the temperature inversion and the PBL developed, as observed during

this haze episode by Wang et al. (2019), the BC aloft could become entrained into the PBL to heat the surface layer and help promote buoyant turbulence and the dissipation of pollutants (Figure 11). This mechanism could have strong influences for policy and we would therefore recommend that further research be performed to directly investigate the mechanism and its potential to influence the severity and longevity of haze episodes.

## 5    Conclusions

Overall, this work showcases and quantifies the various contrasting impacts that BC can play in the aerosol-PBL feedback and consequently the enhancement or dissipation of pollution episodes over Beijing. We also show that there are several factors which may influence both the magnitude and type of effect, including the altitude of the aerosol layers and initial meteorological conditions.

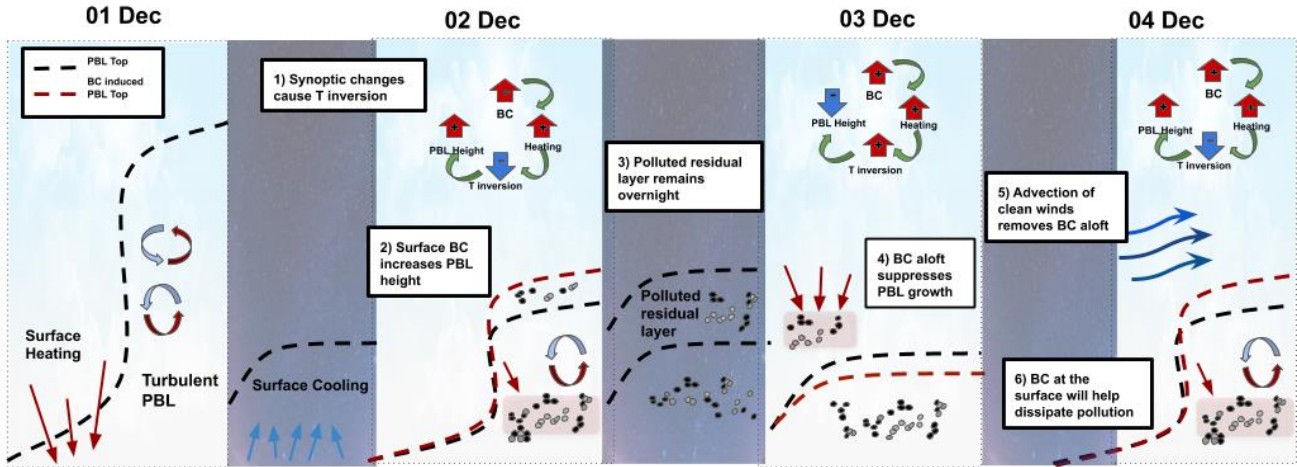

**Figure 11.** Schematic to show the potential impact of BC on a Beijing haze episode which occurred between 1st and 4th Dec 2016, where both the onset and dissipation of the pollution episode is brought about by changing synoptic patterns. The black dashed lines indicate PBL top with the red dashed lines indicating the change caused by BC interactions and the type of impact is dependent on the altitude of the BC layer.

From this work, we suggest: a) The impact of BC aloft on PBL suppression is dependent on the altitude of the aerosol layer in relation to PBL height, b) BC surface heating impact on PBL development is dependent on the strength of the initial temperature inversion and c) When BC is present throughout the column the strong interactions aloft eclipse the impact of BC surface heating. Overall, we show that BC causes heating at a rate of 0.15-0.2 K/h during the daytime which suggests that BC direct radiative forcing has implications for climate, through warming the atmosphere. In terms of the local effect of BC on the aerosol-PBL feedback, BC high above the PBL has little impact on PBL development, while BC just above the PBL suppresses PBL growth. Furthermore, controlling regional emissions of BC will significantly reduce the amount of BC aloft which may reduce the severity of pollution episodes associated with atmospheric stagnation in Beijing.

Here, as shown in Fig. 11, we propose a novel mechanism on how BC may impact pollution episodes in Beijing:

1. At the beginning of the haze episode, synoptic conditions lead to a temperature inversion over Beijing and light winds at the surface, allowing for pollution accumulation in a shallow PBL.

2. BC emitted locally and regionally will be trapped in the surface layer and our results show that this will heat the air at the surface

3. If the temperature inversion is weak (1-3 K) then this will cause sufficient heating at the aerosol loads often found in Beijing to break the temperature inversion at PBL top and enhance vertical mixing to move BC higher in the layer.

4. As the PBL collapses overnight, BC concentrations will exist in residual layers above the nocturnal temperature inversion close to the surface.

5. The following day, these BC layers will heat the air above the PBL and due to the higher SW downwelling radiation aloft, will outweigh the effect of surface BC heating to enhance stability in the column and cause the suppression of the PBL. This exacerbates the pollution event on subsequent days.

6. When changes in synoptic conditions cause the pressure system to move away and upper level winds advect in cleaner air the suppression of the PBL by BC heating diminishes. High BC concentrations in the PBL remain but as we have shown these act to heat the surface and will aid the recovery of the PBL and lessen impact of pollution by promoting mixing at PBL top as the temperature inversion strength weakens (Figure 11).

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
