# Peer review of "The effect of BC on aerosol-boundary layer feedback: Potential implications for urban pollution episodes"

_Atmospheric Chemistry and Physics, 2021_

## Author Response (AR1)

**Response to Reviewer 1**

**As mentioned above, the study is well planned and conducted. However, it seems that finalizing the manuscript lacks some effort. For instance, sometimes sections and figures are incorrectly referred to and the language does not always sound professional. I would also double-check the usage of articles (a/an and the).**

We thank the reviewer for their efforts in going through this work and apologise for the mistakes in the manuscript. These have been checked and corrected in the revised paper. We would also like to highlight a few points in response to both reviewer comments, which we feel may have not been made clear in the original manuscript. Primarily, we would like to highlight that we have used an idealised model framework to examine key processes including the impact of direct aerosol-radiation interactions on turbulent motion. To do this, we have utilised an LES which can resolve turbulent length scales and has two-way coupling between aerosols and radiation. We have applied this modelling framework to the urban area of Beijing during a polluted period as these processes are known to be important in pollution episodes. We have used vertical profiles of meteorology and thermodynamic variables and observed aerosol properties during a polluted episode in Beijing. The key point we wish to highlight to the reviewer is that although the model framework in this case has been applied to Beijing, apart from a few key initial conditions (aerosol properties, meteorological profiles, and surface heat flux values), the model is not specific to Beijing but really is trying to highlight the processes that take place and to examine the impacts of changing certain variables on the aerosol-PBL feedback process. This has now been emphasised upfront in the introduction section of the manuscript as well as in the conclusions.

**In this study, you investigated three different kinds of model setups described in Sections 2.3-2.5. Each setup investigates different model sensitivities. You end up using the word "case" a lot. For instance, you use the word "case" for different setups (Sections 2.3-2.5), but you also use it e.g., in Table 2 ("BC case and No BC case"). Maybe you could try to come up with some more indicate words to make it easier to follow the text? For example, scenario, sensitivity, simulation etc. Different "cases" (i.e., setups described in Sections 2.3-2.5) could also have some more indicative names, e.g., case_aerosol_loading, case_met and case_BC_loading**

This has now been changed in the manuscript - the case names for the three cases have been changed as follows Case 1 – Aero_load, Case 2 – Met, Case 3 – BC_load

The specific simulations have also been given specific names to help with the clarity of the text. The case names used are detailed in the table below. The experimental setup section of the text (Section 2) now also contains a table for each case giving a brief outline of the differences between each of the simulations (Table 3-5).

| Case 1 | Case 2 | Case 3 |
|---|---|---|
| Aero_load_BCsurf | Met_0212_noaero | BC_load_noBC |
| Aero_load_noBCsurf | Met_0212_BCsurf | BC_load_500 |
| Aero_load_BC500 | Met_0312_noaero | BC_load_1000 |
| Aero_load_noBC500 | Met_0312_BCsurf | BC_load_full |
| Aero_load_BC700 | | |
| Aero_load_noBC700 | | |

**The Discussion section now contains conclusions and most of the content in Conclusions should be moved to Discussions. Please review the content of these sections. For instance, Conclusions should not introduce any new arguments, while now the novel mechanism is presented for the first time there**

*These sections have now been changed to reflect an appropriate change in the outline of the manuscript with the novel mechanism and diagram now placed in the discussion section.*

**Limitations of the study have not been discussed anywhere**

A paragraph on the limitations of the study have now been added to the manuscript:

*Our study is an idealised examination of the relationship between aerosol absorption, dynamics and radiation in an urban environment. As a result, the study is not able to fully account for changes in synoptic conditions or understand the impact of regional transport of aerosols on the pollution episodes is a major limitation of the work presented here. Similarly, this work does not look at the impact of secondary aerosol formation which has been found to be a major factor in the rapid increase in PM2.5 concentrations during Beijing haze episodes. Specifically, with regards to the effect of black carbon on aerosol-boundary layer feedback, a current limitation of the work presented here is that it doesn't account for the absorption enhancement of black carbon by scattering aerosols through the lensing effect (Liu et al. 2017). Furthermore, in this work we consider the only absorbing aerosol to be black carbon, while brown carbon has been found to be an important absorber of radiation in several polluted megacities (Beijing and Delhi). Although, BrC isn't as strong an absorber of radiation as black carbon, its presence in high concentrations in polluted urban environments means its impact on these feedbacks should not be discounted. There is scope within UCLALES-SALSA to change the refractive indices and mixing type to reflect some features related to changes in absorption. However, in this work, for simplicity and to allow for the ability to isolate different effects these were not considered.*

**Applying LES in this type of study is very novel and I think you should stress more in the text.**

We have now added the following to the Discussions section of the manuscript to emphasise this:

*Using a coupled aerosol-radiation LES model in this study allowed for direct investigation and quantification of the impact of absorbing aerosols on boundary layer dynamics. There is an array of benefits for using a high-resolution model which directly calculates rather than parameterises turbulent fluxes, for the investigation of heavy pollution episodes. As previously highlighted in this paper, the importance of aerosol-boundary layer feedbacks on heavy pollution episodes, particularly in the megacity of Beijing has been made clear in the literature over the last decade. Primarily, these studies have utilised measurements of both aerosol concentrations, compositions and vertical profiles alongside measurements of boundary layer height and other indicators of turbulent motion such as calculations of sensible heat fluxes, to infer the impact and relationship between aerosol concentrations and properties to boundary layer dynamics. Modelling studies of the aerosol-boundary layer feedback mechanism have mostly utilised regional models such as WRF-CHEM which do not directly resolve turbulent flows. Therefore, the work presented here showcases a novel methodology for investigation of the contrasting impact of absorbing aerosols on boundary layer dynamics and the usefulness of employing such high resolution eddy resolving coupled aerosol-dynamic models to examine physical processes and interactions which can severely influence pollution episodes.*

**Furthermore, visualising the simulation setup would be very useful for the readers. Now you are only showing one-dimensional vertical profiles while LES resolves the three-dimensional flow and concentration fields**

This paper showcases a series of idealised simulations. In all simulations there is no surface heterogeneity or changes in vertical structure across the model field. Due to the lack of heterogeneity across the model field, all results presented, and plots show horizontal domain averages to explain the driving processes as a function of time.

**Specific Comments**

All the specific comments such as typographical and grammatical errors have been addressed directly in the text. Some key responses are highlighted below

**P3 Fig. 1: This figure nicely illustrates the concept of BC aloft and surface BC. However, I do not think it shows the effect of BC layer height on PBL interactions as said in the caption**

The caption has been adapted to now read:

*Schematic showing some of the sources of BC in Beijing, which include industrial emissions, regional and local biomass burning and emissions from transport. As well as outlining the main concepts presented in this paper of the influence of BC aloft and BC within the PBL on PBL dynamics*

**P5 L135-P6 L142: Overall, this paragraph is difficult to follow as the reader is not yet familiar with different "cases"**

The following has now been added to the text:

*Individual cases used to examine the effects of black carbon in this paper are detailed in section 2.2. Overall, three case study experiments with a total of 14 simulations were performed to examine the different impact of: 1) Aerosol loading at different altitudes both with and without the effect of BC (Case Aero_load), 2) Different initial meteorological conditions (Case Met) and 3) Changing concentrations of BC within the aerosol column (Case BC_load). Section 2.2.1 outlines the setup of simulations for the first case study (Case Aero_load) which examines the impact of varying the composition of aerosol layers at different altitudes, including and excluding BC. These six simulations are varied so that there are three different altitudes for an aerosol layer and each layer either has a fractional composition of 10 % BC or no BC (Table 2). In these simulations, the aerosols are only present within the specified layer, with no aerosols present initially above or below the layer. Section 2.2.2 outlines the setup for the second case study (Case Met) which focuses on examining the effect of the initial meteorological conditions on the impact of BC heating within the PBL on boundary layer structure. For these four simulations only a surface aerosol layer is considered and the initial meteorological conditions are either taken from the morning of 02 Dec or 03 Dec 2016. Section 2.2.3 describes the setup for the third case study (Case BC_load) simulations which examine the impact of varying the fraction of BC in different vertical layers for simulations where aerosols are present throughout the column.*

**P3 Fig. 1: This figure nicely illustrates the concept of BC aloft and surface BC. However, I do not think it shows the effect of BC layer height on PBL interactions as said in the caption**

The caption has now been changed to read:

*Schematic showing some of the sources of BC in Beijing, which include industrial emissions, regional and local biomass burning and emissions from transport and an outlie of how BC can interact with radiation both at the surface and aloft to influence PBL dynamics*

**Section 2.1: I would mention somewhere that LES resolves the three-dimensional turbulent field of wind and scalar concentrations and that it directly resolves most of the energy and parametrises only the smallest scales.**

This has now been added to the methodology section.

**P5 Section 2.2: Could you add an illustration of the modelling domain and add the location of the sounding station to that?**

This could be done, but as mentioned in a previous response, all simulations are very much idealised with respect to the surface and the modelling domain. As such, the same initial vertical profiles of meteorological variables are used across the horizontal domain. Including a figure with the modelling domain and position of the sounding station we don't think would add much content to the current manuscript. We have now highlighted in the text the point about the idealised nature of the simulation to avoid confusion.

**P11 L225: Can you further explain this? Above you say that PBL is 4.2 % lower when the BC layer is at 700-1150m compared to the BC layer at 500-950m.**

We recognise this sentence was confusing and has now been replaced in the text to better explain the key points:

The higher aerosol layer (700-1150 m) has less of an impact on PBL height than simulations with the lower aerosol layer (500-950 m). For example, Table 6 shows that case Aero_load_BC500 reduces maximum PBL height by 6.7 % compared to the base case whereas case Aero_load_BC700 only reduces maximum PBL height by 2.96 %. When there are aerosols at 500m, aerosols can become entrained into the upper PBL, as the PBL develops. This results in a strong heating at the top of and above the PBL, causing a decrease the larger decrease in PBL height compared to when the aerosol layer exists higher aloft.

**P13 Fig. 8: How is this vertical integral of TKE calculated? Where do the units kg/s come from?**

This is a mistake in the manuscript and the units should be $kg/s^2$. The vertical integral of TKE is calculated at the total column of TKE. So vTKE = sum(TKE*rho*dz) where rho is density ($kg/m^3$), TKE is turbulent kinetic energy ($m^2/s^2$) and dz is the change in altitude (m).

**P18 L345: Should it be explained somewhere how a night-time stable boundary layer is formed? The word "collapsing" might be misleading for someone who is not familiar with PBL dynamics.**

The term collapsing has now been removed and we have added a sentence on the formation of the night-time boundary layer.

**P19 L368: The illustration in Fig. 11 is helpful but you have referred to it only here and then in Conclusions. I would use it to support the text in the Discussion.**

As was highlighted by the reviewer in an earlier comment, we have edited both the discussion and conclusions sections and have now discussed figure 11 in the discussions sections of the text.

**P20 L381: "saddle type pressure field" has not been mentioned in the text before this.**

We thank the reviewer for recognising this discrepancy, as this hasn't been previously mentioned in this manuscript (it is well described in the paper by Slater et. al (2021) and Wang et. al (2019)) it has been deleted here to avoid confusion.

**Response to Reviewer 2**

We thank the reviewer for their time in reviewing the manuscript and for their insightful comments highlighted below. Specifically, we thank the reviewer for highlighting the recent manuscript by Ma et. al which is very insightful and works well to support the study presented here. Below we have responded to specific comments, technical comments have been addressed or corrected within the manuscript directly.

**There have been researches emphasized the sensitivity of BC impact on PBL due to the BC positions relative to PBLH, which have not been viewed by this study, eg. Yongjing Ma; Jianhuai Ye; Jinyuan Xin\*; et. al (2020). The Stove, Dome, and Umbrella Effects of Atmospheric Aerosol on the Development of the Planetary Boundary Layer in Hazy Region. Geophysical Research Letters, 47, e2020GL087373. I wonder that any new findings in this study compare to previous studies.**

There are numerous findings in this study which relate to the work presented here. We have added the following text to the discussion section to highlight this.

*A recent study by Ma et al. 2020 utilises an LES model to investigate the impacts of the altitude of absorbing and scattering aerosol layers on planetary boundary layer height. Although, in their work they do not specifically treat the aerosol population but focus on changing the aerosol optical depth and aerosol optical properties directly. Their work agrees with the work published here, to show that absorbing aerosols above the PBL will suppress turbulence and PBL growth due to the so-called "the dome effect". Their results also show an increase in PBL height due to absorbing aerosols at the surface promoting turbulent motion, which they term "the stove effect". In contrast to the work presented here, their work shows a much greater increase in PBL height due to absorbing aerosols within the PBL, however, the PM2.5 concentrations over the period investigated by Ma et al. are much lower than in this work. Similarly, the maximum PBL height in that work (around 1000 m) is much higher than in our work. Consequently, it is likely that the specific meteorological conditions are different in Ma et al. and this study and that, as highlighted in section 3.2 of this paper, this has a significant impact on the ability of absorbing aerosols to increase PBL height or the magnitude of "the stove effect".*

**(LES)-aerosol-radiation model has been seldom used in this kind of 3-days long lasting events. I strongly suggest authors to present the detailed results of the LES simulations and the comparisons to observational meteorological factors of PBL and BC and BC related air pollutants profiles in PBL. It is the base and could be the advantage of this study. In the title and content of the paper, I feel the "Potential implications for Beijing haze episodes" could not give a direct and effective cognition to reader. If the mechanism is correct, it should be implicated to the BC related air pollution all over world. And this study did not show the distributions and profiles of BC and air pollutants in Beijing.**

The focus of this work was not to compare specifically to pollution episodes in Beijing but rather to examine various sensitivities relating to black carbon and its influence on planetary boundary layer dynamics. The model setup that was used here has previously been tested and compared to observations (including meteorology) and turbulent metrics (SHF and PBL height) in the previous papers published by these authors ( https://doi.org/10.5194/acp-20-11893-2020 and https://doi.org/10.1039/d0fd00085j). As highlighted in the response to the other reviewer, we feel

that there may have been some confusion about the aim and key features of this work, likely due to these points not being highlighted by the authors in the manuscript. We have therefore added a section in both the introduction and conclusions section of the manuscript to identify this. A key point to highlight is that this study utilises an idealised model framework, with no surface heterogeneity or synoptic drivers but rather examines various processes which are believed to be important for Beijing pollution episodes. Overall, the point of this manuscript is to examine the effects of black carbon on atmospheric dynamics and its feedback on pollution via absorption of radiation using an idealised model framework that can resolve turbulence and has two coupling between aerosols and radiation. We have used measurement data from Beijing to provide input constraint to the model but our findings are generalizable to other polluted environments. We therefore recognise that the title of the manuscript is confusing. This has now been changed to read 'Potential implications for urban pollution episodes'.

**I don't understand the setting of the sensitivity experiments. In section 2, it looks like that the settings were a serial of ideal sensitivity experiments of BC profiles, while in section 3, there were PBL changes in the specific days. When there were no observed BC profiles, we cannot know which setting was close to real profiles of PBL structures. If they are ideal experiments, why did you indicate the results were the specific days in Beijing?**

The settings in terms of the variations in aerosol compositions, vertical profiles and concentrations are varied in these various sensitivity experiments. However, initial meteorological conditions (such as surface temperature, vertical profiles of potential temperature, humidity and wind, which greatly influence the development of the PBL and various moisture and heat fluxes in the model are taken from observations on these specific days in Beijing where a heavy pollution episode took place. LES models, such as the one used in this work are not able to initiate pollution events and no emissions were included in these simulations. Rather, the point of this work is to examine the feedbacks which occur during the pollution episode. Hence, we have used initial meteorological and thermodynamic profiles from observations made during a pollution event to represent typical conditions when high pollution is observed.

**In figure 6, what are the reasons of the BC fluctuations in vertical direction and peak at around 300-500 in figure 6b-d?**

Figure 6 relates to potential temperature lapse rates not BC concentrations.

**In line 305, "this can further increase surface concentrations if the aerosols at PBL top mix down to the surface." correspond to which figure?**

*This was more of a hypothetical statement – figure 5 shows the aerosol layer moving to lower layers around the top of the PBL. This could potentially result in downward mixing of the aerosol layer to the surface but as the simulations presented in this work don't directly showcase this, this sentence has been deleted to avoid confusion.*

**In line 381, "lead to a saddle type pressure field over the region which leads to a temperature inversion" correspond to which figure? If did not discuss in the manuscript, you cannot obtain the conclusion.**

*Again, we thank the reviewer for recognising this discrepancy, as this hasn't been previously mentioned in this manuscript (it is well described in the paper by Slater et. al (2021) and Wang et. al (2019)) it has been deleted here to avoid confusion.*

**Line 171, " the same as BC surface", what is it mean?**

*This is related to the previous case but has now been better clarified in the text through the use of specific case names for each simulation*

**In equation 1, what is the interval of t+1 and t? Is the eq. 1 right?**

*Here t is related to the time (t) at which the SW heating rate is calculated and t+1 is the timestep (2 minutes in these simulations) this has now been clarified in the text.*

**Line 217 "There is slight subsidence of the aerosol layers", ??**

*Figure 5 shows that there is a slight sinking of the aerosol layers which exist only aloft. This is due to the model setup which accounts for large scale subsidence. This has been deleted to avoid confusion.*